# 3D-Printed Conductive Carbon-Infused Thermoplastic Polyurethane

**DOI:** 10.3390/polym12061224

**Published:** 2020-05-27

**Authors:** Namsoo Peter Kim

**Affiliations:** Department of Metallurgical, Materials and Biomedical Engineering (MMBME), Center for Printable Materials Certificate (CPMC), The University of Texas at El Paso, El Paso, TX 79968, USA; nkim@utep.edu; Tel.: +1-915-747-7996

**Keywords:** thermoplastic polyurethane (TPU), carbon-infused TPU, flexible electronics, mechanical property, conductivity, biocompatibility

## Abstract

3D printable, flexible, and conductive composites are prepared by incorporating a thermoplastic elastomer and electrically conductive carbon fillers. The advantageous printability, workability, chemical resistance, electrical conductivity, and biocompatibility components allowed for an enabling of 3D-printed electronics, electromagnetic interference (EMI) shielding, static elimination, and biomedical sensors. Carbon-infused thermoplastic polyurethane (C/TPU) composites have been demonstrated to possess right-strained sensing abilities and are the candidate in fields such as smart textiles and biomedical sensing. Flexible and conductive composites were prepared by a mechanical blending of biocompatible TPU and carbons. 3D structures that exhibit mechanical flexibility and electric conductivity were successfully printed. Three different types of C/TPU composites, carbon nanotube (CNT), carbon black (CCB), and graphite (G) were prepared with differentiating sizes and composition of filaments. The conductivity of TPU/CNT and TPU/CCB composite filaments increased rapidly when the loading amount of carbon fillers exceeded the filtration threshold of 8%–10% weight. Biocompatible G did not form a conductive pathway in the TPU; resistance to indentation deformation of the TPU matrix was maintained by weight by 40%. Adding a carbon material to the TPU improved the mechanical properties of the composites, and carbon fillers could improve electrical conductivity without losing biocompatibility. For the practical use of the manufactured filaments, optimal printing parameters were determined, and an FDM printing condition was adjusted. Through this process, a variety of soft 3D-printed C/TPU structures exhibiting flexible and robust features were built and tested to investigate the performance of the possible application of 3D-printed electronics and medical scaffolds.

## 1. Introduction

The term additive manufacturing (AM) or 3D printing originally referred to a process in which binder material is deposited onto a powder bed layer by layer. The AM refers to a broader sense of different methods, including 3D printing, binder jetting, material extrusion, and the roll to roll process [1,2,3]. The initial development of AM was carried out as research on the subject focused on improving and testing the mechanical and physical properties of mono-dispersed material from the standpoint of printable materials [4,5,6]. Layer-by-layer manufacturing comprises different techniques in which a model is recreated with the help of computer-generated or assisted imaging software (CAD) and rapid prototyping that involves solid freeform fabrication (SFF) of design. FDM involves TPU, which can be durable elastomers or rigid plastics and can be processed using extrusion, injection molding, film blowing, solution dipping, and two-part liquid molding. Threads of metals or polymers, in liquid or paste form, are used to make conductive wires for electric devices [7]. Conductive paste interconnections for 3D electronics were prepared using nanosized copper and oxidation film control [8]. Conductive metal powder packing simulations and methods to improve the conductivity have been reported [9,10]. This technique was applied to fully 3D-printed electronics; however, it still has a limitation in the provided choice of substrates. The studies report on the improvement of mechanical properties in terms of tensile and compression strengths, hardness, and thermal characteristics of the conductive polymer composite (CPC) [11,12,13,14]. The insulating polymer can be convertible to conductive composite through the incorporation of electrically conductive fillers. The development of CPC filaments for the FDM process has commercial potential for printed electrons [15,16,17]. Compared with a metallic conductor, conductive polymer composites have the advantages of ease of shaping, low density, a wide range of electrical conductivity, and corrosion resistance.

Three-dimensional printing can contribute significantly to the field of biomedical applications by producing soft and sturdy 3D-printed structures [18]. The smooth and robust 3D-printed structure has no blood vessels. Without blood vessels, the sleek and healthy 3D-printed structure cannot supply the nutrients required for self-regeneration [19]. The soft and robust 3D-printed structure supplies joints with a low-friction and high resistance exterior. This surface also provides absorption of shock and a high capacity for load-bearing [20]. The articular soft and robust 3D-printed structure present in the knee joint experiences an average load of 3 times the bodyweight of the individual, depending on the activities, and while jumping, the load can be up to 20 times the bodyweight [21].

A composite material comprising TPU and Carbon would be one of the best components to replace this soft, robust 3D-printed structure and sensor. The addition of carbon provides higher mechanical strength to the composite, allowing the end-product to be a flexible yet soft and robust 3D-printed structure scaffold. In addition to the improvement of the mechanical properties of the composite, adding carbon has proven to be biocompatible in vivo [22,23]. 

The demand for 3D-printed scaffolds that are biocompatible, structurally, and mechanically stable in the biomedical research field is increasing over time, and this work contributes to meet this mechanical demand. Although attempts and studies have been made to accurately print hard ceramics, such as artificial teeth or soft materials, such as biomaterials or food, actual in vivo or in vitro tests are expected to require much research [24,25]. For the application of biomaterials, different types of materials can be used for FDM, including metals, plastics, and, more recently, biocompatible materials. As advancements in medicine are made, the demand has grown for new materials to create organ replacements, tissue substitutions, and bone implants, among other biomedical applications [26,27,28].

## 2. Experimental

Flexible and conductive polymer/carbon composite filaments were fabricated for simple and cost-effective production of 3D electronics without nanometal inks or powders. Mechanically blending flexible TPU with carbons, multiwall carbon nanotube (MWCNT), conductive carbon black (CCB), and graphite (G) produced various C/TPU composites. The preparation of filaments of the different composites was adequately completed to obtain 3D-printed functioning electronic devices. The variation of electrical conductivity, hardness, and tensile strength of the composites was measured and interpreted in terms of the type and loading amount of carbon fillers. Prototype structures of 3D-printed electrics were manufactured using a carbon-infused filament and FDM printer. Flexible and conductive 3D structures were constructed using the C/TPU composite filament and a modified do-it-yourself (DIY) style FDM printer.

### 2.1. Filament Preparation

For the synthetic C/TPU composite, TPU was melted first due to its higher melting point of 240 °C. After 20 min, the carbon filler was added to the melted TPU. The composite was continuously stirred for 30 min. The blend was poured on a flat surface at room temperature for the solidification process to take place. During this process, the carbon and C-chain clusters were formed through the distribution or dispersion process. The printability of C/TPU composite filaments by lab-made FDM printers was studied to find the optimum conditions. A 4.0 × 10^−4^ m nozzle was initially attempted, which resulted in clogging of the materials at the nozzle. The diameter of the nozzle was modified to 1.0 × 10^−3^ m. Repeated experiments were performed using the discharge pressure and temperature as variables to obtain stable discharge results under optimal conditions. The structures were printed without defects or clogging and exhibited flexible characteristics. The blending of TPU and three different carbon fillers produced the conductive C/TPU composite filaments: (a) multiwalled CNT made by Carbon Nano-material Technology Co. Ltd. (Seoul, Korea), (b) G from Sigma Aldrich (TX, USA), and (c) CCB, Ketjenblack EC-600JD from AkzoNobel (Tokyo, Japan). Spherical CCB (<35 nm), wire type CNT (diameter: <20 nm, length: <5 μm) and disc type G (diameter: < 1 mm, thickness: <20 μm), were optimized to form a cluster structure and the polydispersity index (PDI) of carbon fillers were maintained in the range of 0.2 to 0.6 for smooth extrusion.

The mixture prepared to solidify on the flat surface was spread across the surface as much as possible, creating a close-to-flat (and as thin as possible) easy to cut sample. The solid mixture was then cut into pieces small enough to be fed into the filament extruder (Figure 1) and repeated 3 times. C/TPU composites were initially produced in pellet form at temperatures in the range of 220–240 °C and then extruded at 200–230 °C at 6 returns per minute (RPM) by a filament extruder which was made by a CPMC specially modified with 1.75 × 10^−3^ m nozzle that obtained the filaments, as depicted in the left image in Figure 1. The extruder consists of a material “feeder,” a motor that rotates to push the material, and a hot end with a temperature and speed control. The pellets of C/TPU were introduced into the feeder. The feeder lays on top of the extruder and needs to be continuously filled. The small pieces of material were then rotated and pushed through a small pipe (1 inch in diameter) into the hot end of the extruder. 

The hot end of the FDM is heated at 200–240 °C (depending on the composition of the blend) and causes the melted materials to be pushed out. The filament is forced out through a hole, which is about 1/16 inches. As the filament is extruded or pushed out, it cools down and solidifies immediately. It needs to be handheld for it to eject uniformly and curl in a manageable way. The quality of the filament is a critical parameter of the printer sample, and it is repeated at least three times under one condition.

### 2.2. Internet of Things (IoT) Enabled 3D Structure Fabrication

3D Printing of C/TPU filament works by continuous addition of thin layers of material. The first step was to download a 3D file from an open-source on the internet, which can also be designed using CAD software (Rhino, USA). The 3D design can then be transferred to Cura (Ultimaker, USA), an open-soft ware program that converts the design into G-code, enabling the printer to interpret it. The program allows for the adjustment of the temperature at which the filament can be melted—the speed at which the melted filament is extruded by controlling the thickness of each layer. The melted filament layers bond to the previous layers as they solidify at room temperature. The quality of the final print is dependent on the temperature, head-speed, and the layer thickness. The disadvantages of this method are a poor surface finish and the fact that it is usually a slow printing technique. The printer was explicitly modified to print the composite filaments with the aim of multi-material printing 3D structures. Six stepping motors with 1.8 degrees step-angle were used to manipulate the X/Y/Z axis and two extruders. The pulleys consisting of 15 teeth and MXL timing belts and pitch sizes of 2.0 × 10^−3^ m were used to control the X and Y-axis motion. Unlike the X/Y axis, the motion of the Z-axis was controlled by threaded rods of 1.0 × 10^−3^ m in diameter with 1.5 × 10^−3^ m pitch size. The stepper unit (SPU) values of the X/Y/Z axis and extruder were 104.984, 104.984, 1066.667, and 132.1131, respectively. Arduino Mega 2560 and RAMPS 1.4 were chosen to control the motion of the 3D printer. Every setting value—number of extruders, default stepper unit for each step motor, type of thermistor, micro-step size of the motor controller—were uploaded to the motherboards by the Marlin Firmware update method. The temperature of the nozzles located at the end of the extruders was monitored through a negative temperature coefficient (NTC) 100 K thermistors. Peculiarly, the G-code created in this way is uploaded directly to the web cloud without being passed to others, and the file is designed to disappear when printing is finished [29]. The IoT-controlled 3D-printed electronic or biosensor printing system can be controlled remotely, without passing the G-code to others. It is particularly advantageous for data containing personal information or specific designs. In this paper, we have introduced a new way to produce C/TPU by combining all of these technologies, successfully separating the team that prepares C/TPU materials remotely from the group that operates equipment and prints through the web-platform. An IoT-enabled teleportation system was introduced. A platform that provides unique design via image, text, and G-code controls the nonface-to-face contact platform (www.makerspiece.com, CPMC, El Paso, TX, USA) [30]. The effect of the 3D-printed C/TPU is to provide a personalized design through a simple “clicks in the web” and plan to build a security design and device.

### 2.3. Electrical Conductivity Test

The FDM process-printed test samples for the electrical conductivity measurements under the printing conditions are listed in Table 1. The dimensions of the test specimens were 63 × 10 × 1 × 10^−3^ m (W × L × T). Four probe measurements were performed on the 3D-printed structures with Loresta-GP MCP-T610 (Mitsubishi Chemical, Tokyo, Japan). Sheet resistivity values were obtained, and volume resistivity and electrical conductivity were calculated in the testing machine. 

### 2.4. Shore Hardness Test

The hardness of C/TPU composite materials was measured with the Shore hardness test with a digital durometer (HOTO instrument, US) with Type-A indenter (flat cone-shaped). Test specimens were prepared by the ASTM D2240-05 Standard Test Method for Rubber Property–Durometer Hardness [31]. Based on this standard, test specimens with dimensions of 15 × 15 × 6 × 10^−3^ m were modeled in a digital CAD (Rhino) and then fabricated with FDM with a constant fill density of 100%, as shown in the left image in Figure 2. The hardness of C/TPU composite materials was measured with the Shore hardness test. This test is typically used as a measure of hardness in polymers, elastomers, and rubbers, so it is suitable for TPU. 

The effect of carbon fillers in the TPU matrix on tensile strength of composites was measured by the tensile test method, ASTM D638-10 Standard Test Method for Tensile Properties of Plastics [32]. The dual column testing system with an extensometer (Instron 5969, Illinois Tool Works Inc., Chicago, IL, USA) was used for the testing, and the speed of the test was kept constant at 50 × 10^−3^ m /min. The FDM printing process with dimensions prepared tensile specimens of type IV for each of the compositions and the pure TPU. Shown in the image to the right in Figure 2, the modified FDM printer with enlarged nozzles and parts for filament feeding was used for printing the C/TPU composite filaments.

## 3. Results and Discussion

### 3.1. Electrical Properties

It is possible to decrease the resistivity of material by dispersing a conducting filler in the matrix. Conductivity measurements of the different concentrations of the fillers used in this work are reported in Figure 3. For both TPU/CNT and TPU/CCB composites, an increase in conductivity was observed with an increased quantity of each carbon filler when the concentration of CNT or CCB was higher than 5.8 wt %. The resistance was too high to measure the electric conductivity of C/TPU when the composition was less than a critical value. The concentration of carbon fillers exceeded 8.0 wt %, and the electrical conductivity significantly increased—behavior, which can be explained by the percolation theory. 

The composite behaves like an insulator when the quantity of conductive particles is below a specific value (percolation threshold, φ_c_). The composite becomes conductive after the percolation threshold value is exceeded. It is believed that a percolation threshold of TPU/CNT and TPU/CCB composites exists in the range of 8–10 wt %. Unlike the TPU/CNT and TPU/CCB composites, the TPU/graphite composite did not show any conductivity increase even when the loading amount of graphite was increased by up to 50 wt %.

The shape and size of the dispersed G particles do not lead to the creation of a conductive path in the TPU matrix because of the aspect ratio of the graphite, and the graphite particles are thin layers with large top and bottom surfaces. Due to the nonpolarity of the G surface, TPU is interposed sufficiently to increase the soft material’s mixing and mechanical strength. Still, for this reason, the contact between graphite is blocked, and the electrical conductivity does not increase. CNT and CCB can be delivered as free electrons, which supports our assumption. These particles were dispersed uniformly in the TPU due to the affinity between the hydrophobic surfaces of the graphite and nonpolar oligomeric ether or ester a chain of TPU. A summary of the composites’ composites and volume resistivity and conductivity measurements for all three composites can be seen in Table 2.

Furthermore, two binary particle distribution systems were also created consisting of TPU75/G20, to which 5 wt % CNT (CNT/G) or 5wt % CCB (CNT/G) was added. In the binary particle distribution systems, there are some changes in the conductivity aspects. The amount of carbon filler was lower than the percolation threshold values in the nanoparticles distribution systems, allowing the binary TPU composite to be conductive. The added CNT and CCB particles increase the probability of connection between the previously separated graphite particles. These results are similar to the reported explanation of others [33,34,35,36]. The predicted conductive path-building mechanism of carbon fillers in TPU is schematically illustrated in Figure 4. In terms of increasing electrical conductivity, CCB is most effective. Still, when CCB is added in the same amount to TPU, tensile strength decreases rapidly, and materials are destroyed. On the other hand, it was confirmed that G does not affect electrical conductivity. However, it does not form a segregated cluster in the TPU and is uniformly dispersed or distributed until 40 wt %. 

In the nanoparticle distribution system, CNT and CCB show that a path formed for conduction to take place; however, for the case of graphite-dispersed parties, the alignment, as well as the high aspect ratio, do not lead to a path for the electron to move. In the case of the binary particle distribution systems, G/CNT, and G/CCB, the diagram illustrates the possibility of electrical conductivity through the formation of a conductive path that is enhanced by the different geometric characteristics of the dispersed particles or distributed clusters.

### 3.2. Shore Hardness and Tensile Test

The Shore hardness of C/TPU composites is indicated in Figure 5. The hardness values initially decreased in all three cases when the filler content was below 8 wt %. This decrease could be due to the weak interface between the carbon particles at higher content of CNT and graphite particles. This could happen because of the enhanced interfacial bonding of filler particles resulting from exceeding the percolation threshold. In contrast, the Shore hardness of TPU continued to be lowered with an increase in the carbon black infill, as noted in the left image in Figure 5. The reason could be the configuration of the carbon black particles. According to their manufacturer, CCB consists of hollow particles with an outer shell, and the composition is 80% porous in volume. Due to the voids in the CCB particles, TPU lost its ability to resist indentation. Young’s modulus in MPa and the S is the Shore A hardness of materials. This equation can provide a more accurate value for S > 40. Young’s modulus is related to the stiffness of materials, so its value has an impact on the ultimate flexibility of the device, as well as on the filament performance before, during, and after printing. The composites exhibited a rapid change in hardness and stiffness when the composition of carbon fillers exceeded their percolation threshold value. At higher content of CNT and graphite, the stiffness of composites increased, while that of the C/TPU composite decreased as the CCB particles increased due to its inner void.

The concentration and type of carbon filler should be appropriately determined by considering the intended use of 3D-printed scaffolds. Previous research on biomaterials—articular soft and robust 3D-printed structures—has helped to understand their mechanical properties, such as tensile strength. The tensile strength of articular soft and robust 3D-printed structures at 20 years of age is an estimated 125 MPa, and at 80 years of age, it is estimated to be around 75 MPa [27]. For use as a biosensor, the elastic modulus of the material can be adjusted within this range. Above all, electrical conductivity and elongation are essential factors in making C/TPU. The percolation theory can be applied by comparing G and CCB. The larger surface of the nonpolar bonding of the G creates an affinity bond in the heterogeneous interfacial C chain of TPU and G, which can maximize entropy mixing by up to 50 wt %. It proves that the electrical conductivity is still close to zero until the G is well dispersed or distributed in TPU.

On the other hand, CCB tends to increase the electrical conductivity even with a small amount of addition to TPU, which is expected to form a segregation cluster. This is because the affinity between CCB particles is higher than that of TPU and CCB. The heterogeneous interface area cannot be increased, resulting in mechanical weakness. The result of the tensile strength (Figure 6) is in good agreement with the theory and Shore hardness test.

Figure 6 demonstrates the stress-strain curves of the various composites tested and digital images of the tested samples after a fracture. TPU with no filler exhibited 40MPa of the ultimate tensile strength (UTS) and elongation of 1940% at fracture, which can be seen in Figure 6a,b. It is confirmed that the addition of carbon particles influences a decrease in tensile strain and lowers the maximum tensile stress for fracture. The UTS was similar in both cases of TPU/CNT composite (23 MPa) and TPU/graphite composite (26 MPa), whereas the maximum elongation of TPU/CNT (970%) was roughly the half of that of TPU/graphite (1565%). There was a definite decrease in tensile stress and strain in the case of the TPU/CCB composite. The maximum tensile stress and strain at break of the TPU/CCB composite were 4.7 MPa and 60.2%, respectively. From Figure 6a,b, the effect of carbon fillers on the stiffness of the composites also was observed. Consistent with the curves, the slope of the curves in the elastic deformation region was differentiated by the type of carbon filler, and TPU/CCB had the highest fragility, followed by TPU/CNT, TPU/graphite, and TPU. 

SEM images of the three carbon fillers used are shown in Figure 7. The mechanical blending of TPU produced the composite filaments. CNT, CCB, and G in conductive TPU/carbon composites with 10 wt % filler were produced in pellet form at temperatures in the range of 493.15 K–513.15 K and were analyzed by SEM. Then, the pellets were extruded at temperatures of 473.15 K to 503.15 K at 6 RPM by a lab-made filament extruder especially modified with a nozzle with a diameter of 1.75 × 10^−3^ m. Elastomers typically show no evidence of necking; that is, the ultimate tensile strength (UTS) of the sample is equivalent to the fracture strength (FS). The fracture mechanism of elastomers can be interpreted in terms of amorphous chains of an elastomer, kinked and heavily cross-linked in the initial stage of tensile testing, and then the chains are stretched but still cross-linked by the application of tensile stress; thus the local strains in the neck are significant in this stage. The cross-links of the chain are ruptured at a certain point (UTS point). 

### 3.3. The Prototype of Soft-Matter Structures

Various compositions of C/TPU composite filaments were printed, and their suitability for commercial FDM printers was tested. Initially, a 4.0 × 10^−4^ m diameter nozzle was used to print C/TPU filaments, but there was a problem with the clogging of the nozzle. To solve this problem, the nozzle was changed from 0.4 × 10^−3^ m to 1.0 × 10^−3^ m in diameter. The printability of the filaments was then retested. Figure 8 shows the fabricated 3D objects using C/TPU. 3D product was made using an optimized C/TPU composites. Two shapes (Figure 8a—hollow cylinder; Figure 8b—(Printing Nano Engineering (PNE) lab logo) were constructed within 20 min, with a total of 50 layers or more. In particular, it evident that the PNE logo was flexible enough to return to the original state after bending numerous times without mechanical defects.

As shown in Figure 9, the 3D structure exhibited a flexible characteristic with electrical conductivity, so it was confirmed that the composite filament is suitable for 3D printing without clogging or defects. This prototype demonstrates the possibility of using C/TPU composite materials in 3D-printed electronics. Figure 9a–c all show excellent electric conductivity flexibility and soft structures. Notably, in Figure 9c, the circuit still works when bent. For more complex devices, the conductivity needs to be improved. Still, there is no doubt that C/TPU composite materials are suitable for printing flexible 3D electronic devices composed of TPU/CNT.

## 4. Conclusions

C/TPU composite filaments were manufactured, and the composites’ resistivity was analyzed to evaluate their feasibility for 3D-printed electronics and biosensors. The structures built with a lab-made FDM printer containing a dual extruding system, enlarged nozzle (0.8–1.0 × 10^−3^ m in diameter), and an advanced filament feeding part. These results imply that C/TPU composites can provide novel ways to build a flexible 3D-printed electronic device as a single structure in a straightforward method. The TPU/CNT composite filament was found to be more stable than the other monodispersed particle filaments by retaining the mechanical properties after printing, specifically up to a composition of 10 wt % CNT. This composite is the best option when considering implants for soft and robust 3D-printed structures in the biomedical area. It was demonstrated that for monodispersed TPU/CNT or TPU/CCB composites, the percolation threshold is in the range of 8–10 wt % of carbon fillers. Although further testing for detailed data on mechanical properties and biocompatibility is still needed, these prototypes demonstrate that C/TPU composite materials have possible applications in the field of 3D-printed or soft and robust 3D-printed scaffolds.

While the composite with dispersed G does not form a sufficient conductive path in the TPU/G mono distribution system, the entropy mixing (ΔSM) of the TPU/G is maximized at 50 wt % of carbon fillers. The flexibility and robustness of TPU/G still maintained up to 50 wt % of G. When CNT or CCB was added to the TPU/G composite, it was confirmed that a soft conductive path in which G was dispersed was formed in the complex. If CNT and CCB fillers were provided, a new concept of a conductive cluster was expected to be produced and confirmed by the electric conductivity test.

## Figures and Tables

**Figure 1 polymers-12-01224-f001:**
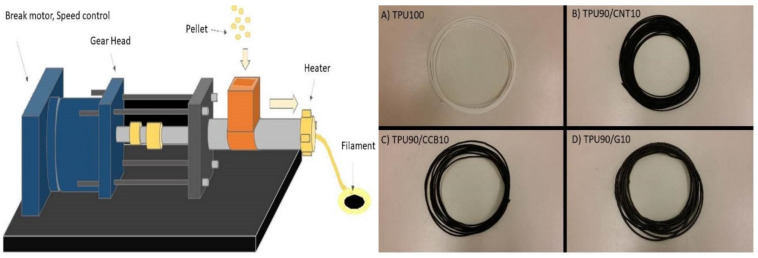
Schematic diagram of the filament extruder (**left**) for mixing the injected carbon black (CCB), carbon nanotube (CNT), graphite (G) with thermoplastic polyurethane (TPU_ and extruded filaments samples; (**A**) TPU100, (**B**) TPU90/CNT10, (**C**) TPU90/CCB10, and (**D**) TPU90/G10 (**right**).

**Figure 2 polymers-12-01224-f002:**
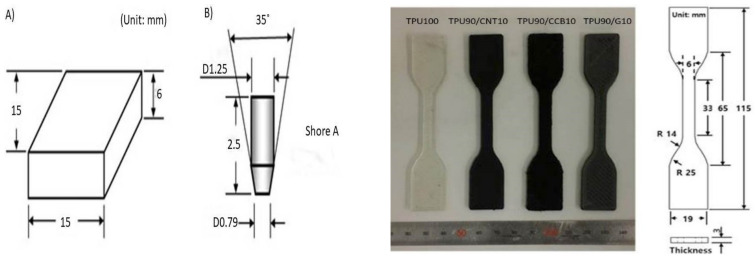
Dimensions of (**A**) Shore hardness test specimen, and (**B**) flat cone-shaped type A indenter (left) and TPU/C composite dog-bone style samples; dimensions of ASTM D638 Type IV tensile test specimens (right).

**Figure 3 polymers-12-01224-f003:**
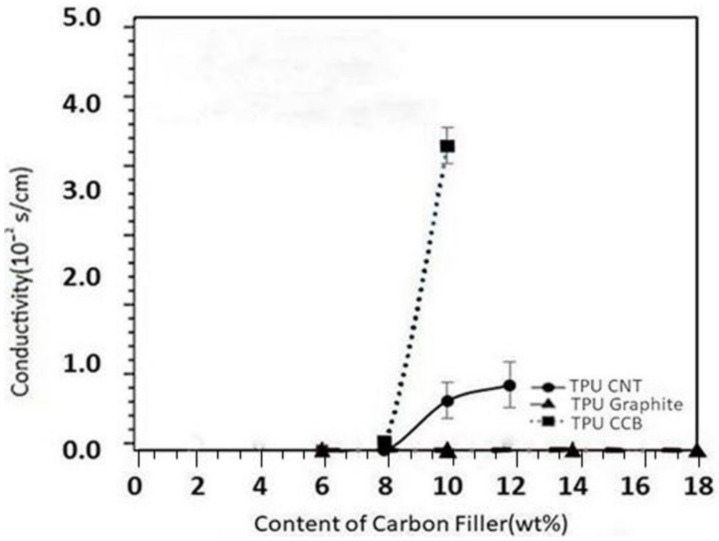
The conductivity of TPU/carbon composites based on the content of carbon filler.

**Figure 4 polymers-12-01224-f004:**
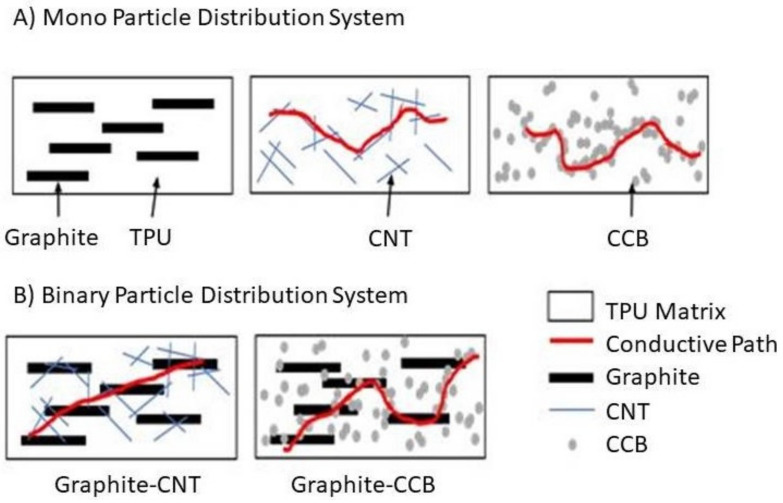
Conceptual diagram of electrical conductivity by particle dispersion or distribution, (**A**) mono-particle dispersion, (**B**) binary particle dispersion.

**Figure 5 polymers-12-01224-f005:**
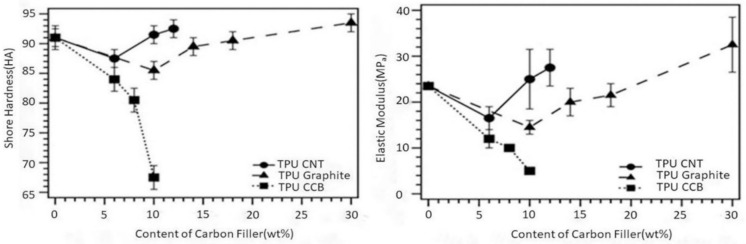
Shore hardness of TPU/C composites (**left**) and elastic modulus of C/TPU composites (**right**).

**Figure 6 polymers-12-01224-f006:**
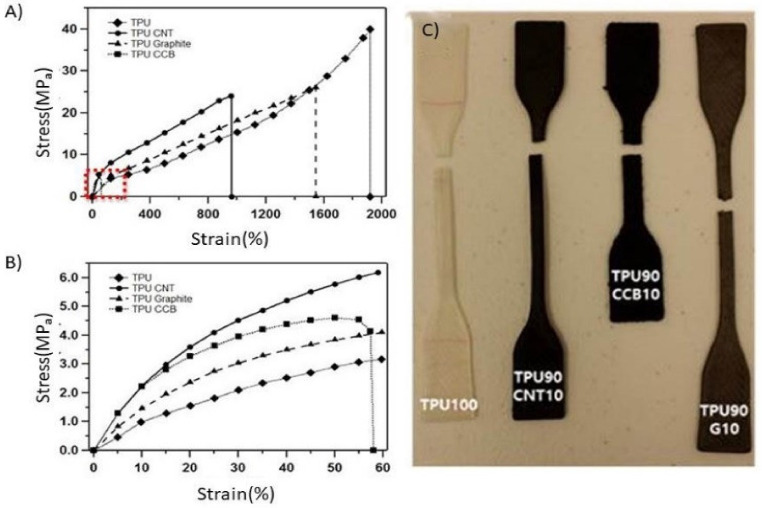
Tensile properties of 3D-printed TPU/C composites objects. (**A**) Stress-strain curve, (**B**) expanded region of stress-strain Curve, and (**C**) specimens fractured under tensile load. The concentration of carbon fillers in all samples in this figure is 10 wt %.

**Figure 7 polymers-12-01224-f007:**
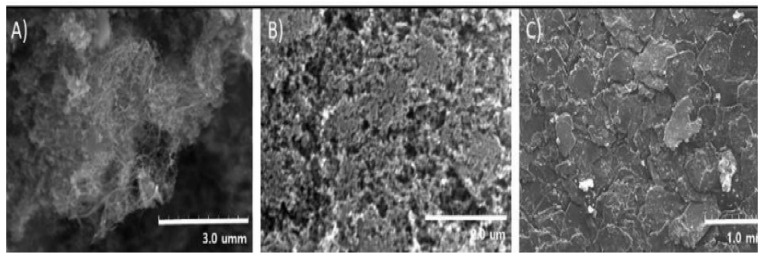
SEM image of conductive carbon fillers. (**A**) Wire-shaped CNT (diameter: <20 nm, length: <5 μm), (**B**) spherical-shaped CCB (<35 nm), and (**C**) disc-shaped graphite.

**Figure 8 polymers-12-01224-f008:**
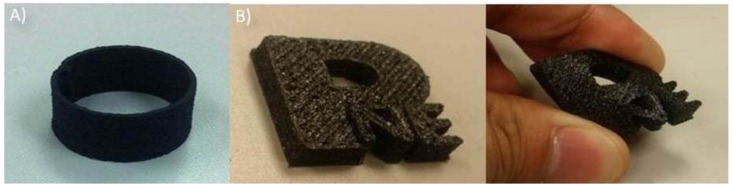
3D product of TPU/CNT (10 wt %) composites. (**A**) A hollow cylinder, and (**B**) (Printing Nano Engineering (PNE)lab logo.

**Figure 9 polymers-12-01224-f009:**
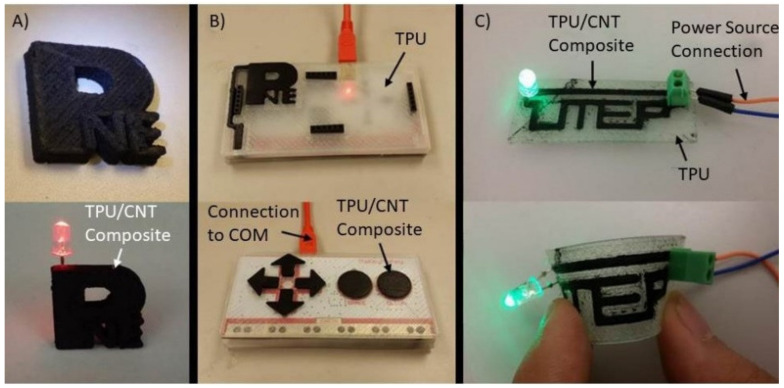
Prototypes of 3D-printed electronic devices made from TPU/CNT (10 wt %); (**A**) PNE lab logo, (**B**) joystick with MakeyMakey Printed Circuit Board(PCB), and (**C**) flexible circuit.

**Table 1 polymers-12-01224-t001:** 3D printing conditions of dog-bone style for the tensile strength test.

Nozzle Diameter (×10^−3^ m)	Print Speed (×10^−3^ m /sec)	Layer Thickness (×10^−3^ m)	Printing Temperature (°C)	Fill Density (%)
1.0	5.0	0.2	230 (±5)	100

**Table 2 polymers-12-01224-t002:** Volume resistivity and conductivity of TPU/CNT and TPU/CCB composites.

Conductive Filler	Composition (wt %)		Volume Resistivity (Ω-cm)	Conductivity (S/cm)	Note
	TPU	Carbon Filler			
**CNT**	94	6	-	-	Nonconductive
	92	8	7.00 × 10^5^	1.44 × 10^−6^	-
	90	10	1.92 × 10^2^	5.77 × 10^−3^	-
	88	12	1.44 × 10^2^	7.58 × 10^−3^	-
CCB	94	6	-	-	Nonconductive
	92	8	1.11 × 10^3^	1.01 × 10^−3^	-
	90	10	8.92 × 10^1^	3.52 × 10^−2^	-
	88	12	-	-	Poor Printability
CNT/G	75	G20/CNT5	4.11 × 10^2^	2.57 × 10^−3^	-
CCB/G	75	G20/CCB5	1.54 × 10^2^	6.69 × 10^−3^	-
^*^ TPU/G composites were not conducive despite the high composition of graphite (over 50 wt %).					

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
