# Peer review of "3D-Printed Conductive Carbon-Infused Thermoplastic Polyurethane"

_polymers, 2020, doi:10.3390/polym12061224_

Round 1

Reviewer 1 Report

In this manuscript the TPU-C composite filaments were manufactured used for FDM technology. It was found that the TPU-CNT composite filament is more stable than the other two fillers by retaining the mechanical properties after printing. For TPU-CNT or TPU-CCB composites, the percolation threshold is 8-10 wt%, while the composite with graphite does not form a sufficient conductive path in the composite. These results demonstrate that C-TPU composite have possible applications for soft electronics or 3D printed scaffolds. However, the scientific content is not so well organized, and the innovation is not described so clearly, in my view, the manuscript should be re-written prior to publication. Here are the comments and suggestions.

  1. The English and format should be improved. Page 3 line 135, the sentence is not well understood.
  2. The Introduction part is too long, which should be cut significantly.
  3. In the experiment section, the brand and the producer of the raw materials like TPU, CNTs, Graphite and CB, as well as the filament extruder company should be given.
  4. Figure 1 and Figure 2 should be combined and reduce the space as they have no scientific content actually. Figure 3 and Figure 4 also should be combined due to the same reason.
  5. In Figure 9, filler contents should be given.
  6. In Figure 10 and Figure 11 the images are not clear. We could not see the fillers structures in the composites despite the content of the nano fillers is very high ~10 wt%. The meaning of the circles drawn inside the images is not clear as the text in the image is illegible
  7. Figure 7 and Figure 8 actually express the same meaning, which should be combined. In this two Figures, it is very strange with CCB incorporation, the modulus decreases significantly. Also for TPU-CNTs and TPU-graphite systems, the firstly decrease and then increase in the modulus or hardness seems not usually.  

Author Response

Comment

Answer to comment

Changes in manuscript

Reviewer 2:

1. Page 3 line 135, the sentence is not well understood.

The author improved the manuscript to reflect reviewers point

See the manuscript Line 77-86

2. The Introduction part is too long, which should be cut significantly.

The author agree on the reviewer’s point that introduction was lengthy and significantly reduce from 93 to 42 lines  

See the manuscript

3. In the experiment section, the brand and the producer of the raw materials like TPU, CNTs, Graphite and CB, as well as the filament extruder company should be given.

We moved the information from Result and discussion to experimental section

See the manuscript Line 97-103

4. Figure 1 and Figure 2 should be combined and reduce the space as they have no scientific content actually. Figure 3 and Figure 4 also should be combined due to the same reason.

We do understand and combined Figs. 1 & 2, and also Figs 3& 4

Please refer to Line 115 for Figure 1 and Line 178 Figure 2.

5. In Figure 9, filler contents should be given.

The author agrees with the reviewer's point.

Please refer to Line 283 for Figure 6

6.In Figure 10 and Figure 11 the images are not clear. We could not see the fillers structures in the composites despite the content of the nano fillers is very high ~10 wt%. The meaning of the circles drawn inside the images is not clear as the text in the image is illegible

The author agrees with the reviewer's point. The clarity of Figure 10 was increased, Figure 11 was removed, and comments were added to reflect the author's comments.

Please refer to Line 295-297 for Figure 7

7. Figure 7 and Figure 8 actually express the same meaning, which should be combined. In this two Figures, it is very strange with CCB incorporation, the modulus decreases significantly. Also for TPU-CNTs and TPU-graphite systems, the firstly decrease and then increase in the modulus or hardness seems not usually. 

The author combined Figures 7 and 8. The CCB used in this paper is a hollow form with 80% porosity, and as the concentration of CCB increases, the physical strength of TPU-CCB is decreasing and consistent with the tensile results below. Besides, in the case of CNT and G, the physical intensity is decreased by the initial formation of the crust rather than the physical strength increase due to the entropy mixing of the initial material and the interface increase by the material and explained in this manuscript line 257-266

Please refer to Line 249-251 for Figure 5 and see the manuscript Line 257-266 in RED

Reviewer 2 Report

Manuscript ID: polymers-809466

Title: 3D Printed Conductive-Carbon Infused Thermoplastic Polyurethane

The paper entitled “3D Printed Conductive-Carbon Infused Thermoplastic Polyurethane” by Namsoo Peter Kim reported an interesting experimental and modelling discussion in the enhanced mechanical-flexibility and electrical conductivity of 3D printed thermoplastic polyurethane (TPU) composite by using three-carbon fillers such as multi-walled carbon nanotube (MWCNT), carbon black (CCB), and graphite (G). The experiments were carefully conducted and the manuscript were very well organized. I recommend this work is suitable for publication in this journal after the following few comments are addressed.

  1. The author used the CNT, CCB, G, and TPU to fabricate composite. However, the author did not supply the fundamental description, such as the molecular weight, polydispersity index (PDI) of the polymer, dimension size of carbon-based materials.
  2. It will be better if the author can observe the well dispersion morphologies of graphite-CNT and graphite-CCB hybrid in TPU matrix for TEM observation. Explanations for the relationship between electrical conductivity and 3D-network dispersion degree are needed.
  3. In Figure 11, the representation is not clear to understand due to low quality and need to be re-drawn. And, fonts in several figures can be made bigger for clarity. The figures need to ensure higher quality.
  4. Overall, it is recommended for publication in the Polymers after the major revisions.

Author Response

Comment

Answer to comment

Changes in manuscript

Reviewer 1:

1.      The author used the CNT, CCB, G, and TPU to fabricate composite. However, the author did not supply the fundamental description, such as the molecular weight, polydispersity index (PDI) of the polymer, dimension size of carbon-based materials.

The author agree on the reviewer’s point and revised the line 96-103

See the manuscript in RED

2.      It will be better if the author can observe the well dispersion morphologies of graphite-CNT and graphite-CCB hybrid in TPU matrix for TEM observation. Explanations for the relationship between electrical conductivity and 3D-network dispersion degree are needed.

According to the reviewer’s comment, TEM images have been analyzed, however, dispersion and distribution carbon filler was not be distinguished in TPU matrix. So, The author improved the manuscript to reflect reviewers' points by thoroughly examining the manuscript and added a more in-depth discussion in line 290-294

See the manuscript in RED

3.      In Figure 11, the representation is not clear to understand due to low quality and need to be re-drawn. And, fonts in several figures can be made bigger for clarity. The figures need to ensure higher quality.

The author agrees with the reviewer's point. The clarity of Figure was increased, Figure 11 was removed, and comments were added to reflect the author's comments.

Please refer to Line 295 for Figure 7

Round 2

Reviewer 1 Report

The authors made good revisions and now it is acceptable

Reviewer 2 Report

The manuscript was revised carefully and improved so much according to reviewers’ suggestions. The scientific insights are expressed well in this manuscript. Overall, the current revision is recommended for publication in the Polymers.